Size class structure, growth rates, and orientation of the central Andean cushion Azorella compacta

Kleier Catherine 1 ckleier@regis.edu
Trenary Tim 2
Graham Eric A. 3
Stenzel William 4
Rundel Philip W. 5
1 Department of Biology, Regis University , Denver, CO , USA
2 Department of Mathematics, Regis University , Denver, CO , USA
3 Nexleaf Analytics , Los Angeles, CA , USA
4 Department of Computer Information Systems, Metropolitan State University , Denver, CO , USA
5 Department of Ecology and Evolutionary Biology, University of California , Los Angeles, CA , USA
Esler Karen
Electronic publication date: 2015 Mar 17
Publication date: 2015
Volume: 3
Electronic Location ID: e843
Received 2014 Nov 19; Accepted 2015 Feb 27
Copyright: © 2015 Kleier et al.
Copyright year: 2015
Copyright holder: Kleier et al.
License: This is an open access article distributed under the terms of the Creative Commons Attribution License, which permits unrestricted use, distribution, reproduction and adaptation in any medium and for any purpose provided that it is properly attributed. For attribution, the original author(s), title, publication source (PeerJ) and either DOI or URL of the article must be cited.
License URL: https://creativecommons.org/licenses/by/4.0/

Keywords: Andes, Parque Nacional Lauca, Growth rate, Cushion plant, Puna

Funding: National Geographic Waitt Foundation W154-11 This work was funded by a National Geographic Waitt Foundation grant W154-11. The funders had no role in study design, data collection and analysis, decision to publish, or preparation of the manuscript.

==============================
Azorella compacta (llareta; Apiaceae) forms dense, woody, cushions and characterizes the high elevation rocky slopes of the central Andean Altiplano. Field studies of an elevational gradient of A. compacta within Lauca National Park in northern Chile found a reverse J-shape distribution of size classes of individuals with abundant small plants at all elevations. A new elevational limit for A. compacta was established at 5,250 m. A series of cushions marked 14 years earlier showed either slight shrinkage or small degrees of growth up to 2.2 cm yr−1. Despite their irregularity in growth, cushions of A. compacta show a strong orientation, centered on a north-facing aspect and angle of about 20° from horizontal. This exposure to maximize solar irradiance closely matches previous observations of a population favoring north-facing slopes at a similar angle. Populations of A. compacta appear to be stable, or even expanding, with young plants abundant.

Introduction

Azorella compacta (Apiaceae), a large woody cushion plant, forms an iconic species of the Altiplano Plateau of northern Chile, Bolivia, Argentina, and Peru (Kleier & Rundel, 2004). Known locally as llareta, it forms broad irregular cushions that commonly reach diameters of 3–4 m, or much more, on rocky slopes at high elevations. Its range extends across the Altiplano Plateau of the south-central Andes from southern Peru through western Bolivia and into the northeastern Chile and northwestern Argentina (Martinez, 1993). The species only rarely occurs below 4,000 m and an upper elevational limit of 5,200 m has been reported (Halloy, 2002), making it one of the highest occurring woody plant species in the world.

Azorella compacta forms unusual bright green woody mounds on steep rocky slopes where few other plants survive (Fig. 1A). Thousands of small stems grow so tightly together that the plant’s surface has the consistency of smooth, green wood (Fig. 1B). Concerns about the conservation of this species due to past major harvesting for fuel in the early and mid-20th century, has caused A. compacta to be classified as a “data deficient” species (IUCN, 2012). Previous research on A. compacta has shown that these plants are found more frequently on the north side of large boulders on north-facing, rocky slopes and that smallest size classes were most frequent (Kleier & Rundel, 2004). Further ecophysiological work with a model showed that A. compacta would have increased radiation interception on north-facing slopes and that A. compacta could have a surface temperature 10 °C warmer at dawn than a non-cushion forming co-occurring mat plant (Kleier & Rundel, 2009).

Figure 1 Azorella compacta photographs.

Azorella compacta. (A) irregular cushion form of growth (B) surface of male cushion.

The present research continues a long-term study of A. compacta begun in 1998, and expands existing data for growth rates in these cushions by revisiting plants marked 14 years earlier. A second objective was to sample a larger elevation gradient of A. compacta populations, extending from 4,400 to 5,250 m, to determine the presence of correlations with density, size, or elevation, and to determine if these correlations might show an upslope niche shift indicative of response to climate change (Lenoir et al., 2008). Finally, previous investigations on the significance of energy balance in cushion establishment (Kleier & Rundel, 2009) were expanded to look at aspects of orientation and solar irradiance in mature cushions themselves and not only the slope face.

Methods

Site description

Field studies were carried out in Lauca National Park, a protected area located 145 km east of the coastal city of Arica and adjacent to both Peru and Bolivia. The park covers 1,379 km2 of land classified as the central Andean dry puna (McGinley, 2009), with elevations ranging from 3,220 to 6,342 m. A UNESCO World Heritage site, the park is renowned for high-altitude lakes Chungará and Cotacotani, and a rich diversity of wildlife and flora (Rundel & Palma, 2000). Rainfall averages 320 mm annually, with three-quarters falling during the summer, January through March. Mean air temperatures at 4,400 m reach 20–25 °C during the day and fall below freezing at night in all but 2 months of the year (Rundel & Palma, 2000). The broad Altiplano Plateau in Lauca National Park lies largely at elevations of 4,400–4,900 m elevation, but with higher volcanic slopes, which are home to extensive populations of A. compacta.

The Andean Cordillera in the study region consists of folded and faulted Cretaceous and Tertiary sediments mixed with former volcanic centers of activity. Much of the substrate geology in the study region is formed by a chain of deeply eroded Miocene volcanoes, which make up the western margin of the Lauca Basin, and which are sometimes termed the Chilean Western Cordillera. The most prominent peaks are the Nevados de Putre (5,775 m) and Cerro Belén, Cerro Tallacollo, Cerro Orotunco, and Cerro de Anocarire all of which reach above 5,000 m. Several relatively young volcanic cones rise above the Altiplano plateau, including the Parinacota (6,342 m), Ponerape (6,240 m), and Guallatire (6,063 m) within Lauca National Park (Rundel & Palma, 2000).

Size class structure and elevational range

To assess elevational gradients in population structure, we measured 406 cushions sampled in 30 separate 100 m line transects established on rocky slopes with A. compacta populations throughout Lauca National Park. The lowest elevation transect was 4,247 m and the highest was 5,182 m. There was at least 500 m between the beginning of each transect. The line intercept of each A. compacta cushion along these transects was recorded to the nearest cm. Each cushion was measured along two orthogonal axes, roughly corresponding to the greatest width and length, to provide a squared estimate of surface area (Kleier & Rundel, 2004). The tape measure was allowed to follow the surface of the plant to account for irregular planar features. This was necessary as some plants have more undulations within them than others. GPS measurements were made to record the latitude, longitude, and elevation at the beginning point of each transect.

Elevational transects were extended on two different peaks, an unnamed peak that Corporación Nacional Forestal de Chile (CONAF) rangers called Cerro Apacheta Choquelimpie (5,289 m) and Cerro Larancaugua (5,447 m), to visually search for the highest occurring individual of A. compacta. Access was restricted by heavy snow and ice cover and avalanche risk to two higher peaks: Volcán Parinacota and Volcán Pomerape.

Determination of growth rate

The growth rate of A. compacta was determined by changes in dimensions of marked individuals that were first tagged in 1998, measured again in 2000 (Kleier & Rundel, 2004), and resampled in January 2012. These plants are located approximately 2.5 km northwest of the village of Parinacota along the path to Lagunas Cotacotani (18°12.554′S and 69°16.132′W) at an elevation of 4,454 m. Although 100 plants in four separate plots were originally marked, only 9 of the marked plants within one plot were able to be relocated. Presumably, A. compacta completely grew over at least some of the permanent tags of the remaining plants. However, one plot of tags was removed between 1998 and 2000, and it is likely that more were removed between 2000 and 2012, due in part to controversies regarding ownership and control of park land. In 1998, park staff indicated that the proposed plots would be located on public land. However, in 2012 we found several painted messages denoting the area as private property. For the nine remaining tagged plants, we measured length and width in orthogonal axes across the apex of the cushion, perimeter, and height, which was determined from the apex of the cushion to the nearest western edge. We also noted any dieback (increase in dead tissue) and the presence of flowers or fruits.

Cushion orientation

The aspect and the angle from horizontal that maximized the projected area of 53 individual A. compacta cushions were determined visually with a compass and clinometer. After an isolated cushion was identified in a flat area without significant influence from local terrain, a raster-like approach was used. The assistant stood approximately 2 m from the individual cushion at a low angle (crouching) and walked in an arc around the plant, visually gauging the projected area at different aspects. When an aspect had been determined that maximized the projected area of the cushion at the low angle, the angle was increased (the assistant stood at an increased height off the ground), and the process was repeated until a maximum projected area was determined for all aspects and angles. A transect line was then used to connect the center of the individual cushion to the point in space that maximized the visual projected area of the cushion and the aspect of that transect line and the angle from horizontal was measured (Fig. 2). The same field assistant was employed for all measurements to avoid changes in bias between individual measurements. The declination from magnetic north of 5.33°W was determined for latitude 18°12′6.70″S, longitude 69°16′5.16″W for January 6, 2012 using the online NOAA Estimated Value of Magnetic Declination Calculator http://www.ngdc.noaa.gov/geomag-web/#declination.

Figure 2 Sampling design for measuring cushion orientation.

Sampling design for measuring cushion angle and azimuth of orientation.

Statistical analysis

For demography data, we used SPSS version 19 (IBM, USA). We used Pearson Correlation to determine if there were more plants at higher elevations and to determine if plants were smaller at higher elevation. We used a Wilcoxan signed rank to determine differences in growth rate because the small sample size meant that the data were non-parametric. We used a Rayleigh uniformity test to detect differences in orientation. These analyses were performed in R version 2.15.1.

Results and Discussion

Size class structure and elevational range

A histogram demonstrates that the smallest size classes of A. compacta are most common at all elevational ranges (Fig. 3). We grouped elevation into three categories: <4,500 m; 4,500–5,000 m; and >5,000 m. These categories represent the lower range of A. compacta (<4,500 m), the range that has the greatest number of plants (4,500–5,000 m), and the upper range (>5,000 m).We used these categories to better illustrate the trend with size class and density with elevation. This trend is the same when plants were measured using perimeter, instead of area. The mean canopy area for the 406 cushions measured, calculated as length × width, was 2.9 m2 (±2.10 SEM). A Pearson correlation analysis found a slight (r = 0.129), but significant (p = 0.009), negative relationship between elevation and size of plants. However, much of this pattern is due to the presence of a number of very large plants in transects sampled above 5,000 m. The number of plants per 100 m transect ranged from 6 to 24, with a mean of 13.5 plants, and the number of plants per transect did not significantly correlate with elevation.

Figure 3 Size class distribution at three elevations for A. compacta.

Relative proportion of cushion sizes in three groups of elevational populations of Azorella compacta.

The large area of many A. compacta cushions is not unique within this genus. Continuous mats of Azorella selago on the subantarctic Marion Island can be tens of meters across (Huntley, 1972), although these broad mats have been shown to often consist of multiple individuals grown together (Mortimer et al., 2008; Cerfonteyn et al., 2011). A similar pattern of merged canopies is likely present in A. compacta. For A. selago, smaller round cushions are found growing at all angles to the slope, but as cushions become larger and more elongated, growth is oriented vertically perpendicular to the plain of steeper slopes (Boelhouwers, Holness & Sumner, 2000). Azorella monantha in the central Andes of Argentina occur as broad carpets that grow over all manner of objects including rocks, debris and other plants (Méndez, 2011).

Size class structure in A. compacta follows the same trend of a reverse J-shaped curve of population distribution that was noted in 2000, with many smaller plants in what appears to be a pre-reproductive stage, i.e., <2 m perimeter (Kleier & Rundel, 2004), and this pattern is repeated at all elevations including those growing above 5,000 m. This population structure suggests that there is some degree of regular success in the establishment of cushion seedlings. There are clearly tradeoffs between seedling establishment and life span in many alpine plants, but many cushion plants seem able to maintain such success as well as great longevity. Similar population structures have been reported for Azorella madreporica in the high Andes of central Chile (Fajardo, Quiroz & Cavieres, 2008), Azorella selago in the subantarctic Indian Ocean (le Roux & McGeoch, 2004), and in the closely related Llaretia acaulis in the Andes of central Chile (Armesto, Arroyo & Villagran, 1980). Cushions of Eritrichium nanum in the Austrian Alps also exhibit a reverse J-shaped curve of population distribution (Zoller & Lenzin, 2004).

Such size-age structure can be readily maintained by episodic but frequent seedling recruitment, followed by relatively low rates of mortality once these seedlings are established (Doak & Morris, 2010). Poor recruitment of seedlings in the temperate alpine cushions Minuartia obtusiloba and Paronychia pulvinata is balanced by an estimated longevity of 200 and 324 years, respectively (Forbis & Doak, 2004). Similarly, the alpine cushion Silene acaulis in the Pyrenees Mountains has irregular seedling establishment but life spans in excess of 300 years (Morris & Doak, 1998; García, Guzmán & Goñi, 2002).

Our field measurements included a new high elevation record for A. compacta at 5,250 m, 50 m higher than previously reported (Halloy, 2002). The individual found at this elevation was not flowering and was of a size <2 m perimeter that may not be reproductive (Kleier & Rundel, 2004). The species almost certainly grows at even higher altitudes on the slopes we surveyed, but a deep snowpack at the time of sampling restricted access. Since cushions were not smaller with increasing elevation, A. compacta are not showing the typical niche shift of moving up in elevation as a response to climate change (Lenoir et al., 2008).

Growth

The changes in perimeter over 14 years for the nine plants that were tagged in 1998 are shown in Table 1. We used a Wilcoxan signed rank test to determine if there was any difference in the perimeter of the plants between 1998 and 2012. The results indicated that the median change was not significantly different from zero over the 14 years, V = 17, p = 0.94.

Table 1 Perimeters of individual Azorella compacta in 1998 and resampled in 2012.

Individual	1998 perimeter (m)	2012 perimeter (m)	Change in perimeter (m)	
1	0.78	0.88	0.1	
2	4.3	4.23	−0.07	
3	6.13	5.23	−0.9	
4	1.15	1.11	−0.04	
5	3.61	3.92	0.31	
6	5.12	5.37	0.25	
7	4.47	4.3	−0.17	
8	8.64	8.25	−0.39	
9	5.51	4.94	−0.57	
Average ± SD	4.41 ± 2.42	4.25 ± 2.24	−0.16 ± 0.39	

Despite these slow rates of mean growth, we also found that individual A. compacta can grow significantly more quickly under some conditions. As an example of rapid growth, we observed a semi-rectangular individual 20 cm by 40 cm, with a perimeter of 110 cm growing in a ditch (Fig. 4) on the side of the Highway 11, which travels from Bolivia to Arica, Chile through Lauca National Park. The ditch was presumably created when the highway was repaved in 1996. Thus, this individual is at most 16 years old and would have a minimum estimated growth in perimeter of 6.88 cm per year. It is possible that young A. compacta may grow more quickly in a planar fashion, while older plants allocate more growth to vertical changes in surface area, though we did not measure this.

Figure 4 Photo of A. compacta in a ditch.

Azorella compacta growing in a ditch alongside Highway 11, which connects La Paz, Bolivia and Arica, Chile. The highway was paved in 1996, presumably when the curb was constructed, and thus this plant is at most 16 years old. Photo was taken January 5, 2012.

Our findings of slow or even negative growth rates for A. compacta are supported by our previous research, which reported a mean radial growth rate of 1.46 cm yr−1 over 14 months (Kleier & Rundel, 2004). The large size and slow rates of growth established for A. compacta clearly indicate a great age of centuries or more for the larger cushions. The current study of growth averaged over 14 years shows slow but variable rates of radial growth of about 0.4 cm yr−1, although this is based on a small sample size (n = 9). This growth rate also includes six plants that shrunk over this time period (Table 1). Salguero-Gómez & Casper (2010) illustrate the need to include plant shrinkage in demographic models.

Other studies have suggested even lower growth rates for A. compacta. Ralph (1978) reported annual radial growth averaging about 1.4 mm yr−1. Halloy (2002) reported average radial growth of 1.55 mm yr−1, but also found that individual plants could grow at rates up to 12.3 mm yr−1, consistent with our observations of faster growth in young plants.

Halloy (2002) also reported that growth in A. compacta is seasonal, reflecting the highly seasonal summer precipitation regime of its habitat. Although the Altiplano climate regime presents favorable daytime temperatures for growth throughout the year, two-thirds of the annual precipitation falls in January and February, with a long dry season from April through November that accounts for only 4% of the total.

Slow rates of radial growth have been reported in other alpine cushion plants. Silene acaulis in the Rocky Mountains which has been reported to have a radial growth rate of 1.0–1.5 cm yr−1 (Benedict, 1989), and the arctic cushion Diapensia lapponica has a mean radial growth rate of only 0.6 mm yr−1 (Molau, 1996). Radial growth rates for cushions of A. monantha in the central Andes of Argentina are 1.15–190 cm per year (Méndez, 2011), while A. selago on sub-Antarctic Marion Island ranged from an average of 0.28 cm per year (Frenot et al., 1993) to 0.426 cm per year (le Roux & McGeoch, 2004).

Our study also indicated that the way plants are measured changes overall growth rate substantially. Several authors have noted that growth in A. monantha mats does not occur at equal rates in all directions (Halloy, 2002; Méndez, 2011), supporting our concerns about the manner in which growth rates should be measured. Unlike growth measurements of the temperate cushion Silene acaulis (Morris & Doak, 1998), A. compacta cannot be measured in simple terms of radial growth because this omits volume of the plant. Likewise, growth measurements of the congener, A. selago, were analyzed by using height (le Roux & McGeoch, 2004), but that is not possible with A. compacta because the cushion is too dense and often forms over small boulders. To take a height measurement, a hole would have to be drilled through the plant. Similar concerns about growth as a function of volume are true for northern populations of A. madreporica and L. acaulis. Thus, further ontogenetic models are necessary to determine more robust growth rates for cushions with large volumes.

Cushion orientation

Despite their seemingly irregular surface, the orientation of A. compacta cushions showed strong patterns favoring the maximum exposure of cushion surface area to annual solar radiation. Of the 53 plants we measured, 60% of the plants had a maximum exposure facing −30–30° from true north, with a mean direction of 8.82°. No plants had a maximum exposure of surface area that was more than 90° from north. A Rayleigh test for uniformity indicated that the distribution of cushion orientation was significantly different from uniform, indicated clustering toward northern exposure (p < 0.001) (Fig. 5). The angle of maximum exposure showed a marked orientation with a mean inclination of about 20° from horizontal. Almost 60% of cushions had an angles between 16 and 30°(Fig. 6).

Figure 5 Distribution of orientation by aspect for A. compacta.

Relative distribution of orientation by aspect of Azorella compacta. Most cushions orient towards the north (equator facing), n = 53.

Figure 6 Distribution of angles from horizontal for Azorella compacta.

Relative distribution of orientation by angle from horizontal cushions of Azorella compacta, n = 53 orientation.

This orientation of exposure and inclination not only maximizes solar irradiance over the course of the year, but dampens the seasonal swings in irradiance that occur on a normal surface. Although an equivalent angle of inclination with a south-facing exposure would add up to 20% greater irradiance in summer, this orientation would receive less than half of the winter irradiance received by the north-facing exposure.

Microsite selection by A. compacta strongly favors establishment at the base of moderate to large-sized boulders, and preferentially on the north-facing side (Kleier & Rundel, 2004). Nowhere in our surveys have we observed individuals growing in sandy soils without boulders present. One potential advantage of such positions would be that heat storage in boulders could provide some benefit to adjacent seedlings in buffering diurnal changes in soil temperature (Poesen & Lavee, 1994). Positions adjacent to boulders may also offer favorable conditions of water availability in arid and semi-arid regions. Boulders can influence surrounding hydrology by collecting surface flow, slowing evaporation caused by soil warming, and condensing moisture in the evening at the rock/soil interface (Flint & Childs, 1984; Nobel, Miller & Graham, 1992; Poesen & Lavee, 1994).

Previous work with A. compacta demonstrated a strong and significant correlation of establishment on the north side of boulders (Kleier & Rundel, 2004). This finding suggested that energy balance may be a more significant factor in microhabitat selection. Azorella compacta cushions, despite their irregular form, strongly favor maximizing exposure of surface area to solar radiation. Cushions are oriented to favor an exposure to the north at angle centered on about 20° from horizontal. Because A. compacta is found at 18°S, well within the Tropic of Capricorn, the dominant orientation found in this study favors solar radiation input. Models of total daily solar irradiance over a seasonal cycle showed that north-facing slopes at a slope angle of 20°, very close to the favored slope angle for A. compacta establishment (Kleier & Rundel, 2009), received more annual irradiance than those on horizontal or south-facing slopes of the same angle.

CONCLUSIONS

Although populations of A. compacta appear to have rebounded well since heavy harvesting of the cushions for fuel in the early and middle parts of the last century, growth rates of individual plants are clearly very slow. Low growth rates for A. compacta led Alliende & Hoffmann (1983) to consider that the species could become threatened under conditions of continued harvesting for fuel. Likewise, Benoit (1989) concluded that the species is vulnerable in Chile. However, we have observed large reproductive populations of A. compacta across Lauca National Park, and little evidence of significant harvesting. Although we did observe very low growth rate, at least in this area, the future survival of A. compacta does not appear to be under significant threats from direct human interactions as size class structure shows many smaller plants. Although global change models predict a 3–4 °C rise in temperatures in the central Andes (Anderson et al., 2011), which might well shift elevational distributions several hundred meters higher, we did not show evidence of smaller plants at higher elevations, but we did support the importance of maximizing solar radiation in terms of plants orienting in northerly directions.

Supplemental Information

Supplemental Information 1 Angle surveys of A. compacta

Dataset of angle and aspect of A. compacta.

Click here for additional data file.

Supplemental Information 2 Size data for A. compacta

Sizes of A. compacta from 1998 to 2012.

Click here for additional data file.

Supplemental Information 3 Transects of A. compacta

Size data for 100 m transects—measuring 406 individual plants.

Click here for additional data file.

We thank two interviewees in Parinacota, Chile, who wish to remain anonymous. Corporacion Nacional Forestal de Chile (CONAF) officials provided important logistic support and necessary permits for this work. Alicia Malet provided valuable editing assistance, and three reviewers provided valuable comments to this manuscript.

Additional Information and Declarations

Competing Interests

Author Contributions

Eric A. Graham is an employee of Nexleaf Analytics.

Catherine Kleier conceived and designed the experiments, performed the experiments, analyzed the data, wrote the paper, prepared figures and/or tables, reviewed drafts of the paper.

Tim Trenary analyzed the data, prepared figures and/or tables, reviewed drafts of the paper.

Eric A. Graham performed the experiments, analyzed the data, contributed reagents/materials/analysis tools, prepared figures and/or tables, reviewed drafts of the paper.

William Stenzel performed the experiments, wrote the paper, reviewed drafts of the paper, helped find the funding opportunity and write initial grant.

Philip W. Rundel conceived and designed the experiments, performed the experiments, analyzed the data, wrote the paper, reviewed drafts of the paper.

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
