# Peer review of "Size class structure, growth rates, and orientation of the central Andean cushion Azorella compacta"

_PeerJ, doi:10.7717/peerj.843_

## Round 0.1 · original submission · Major Revisions

Given the potential value of this work towards our understanding of Andean cushion plants, and the fact that there is a long-term aspect to the work (which is rare), I would like to encourage the authors to resubmit their paper after carefully considering the comments of reviewers 1 and 3. In particular, please attempt to focus the paper more effectively and minimize speculation; address the issues linked to experimental design and address questions of clarity.

Reviewer 1 ·

Basic reporting

I feel the article is not coherent; it has bits on orientation, demography, elevational range and growth making it inappropriately subdivided.

Experimental design

The core of the paper concerns growth estimates. Although 100 plants were marked only 9 were found. This is too few, certainly for 10 000 replications in a boot-strap analysis because growth was highly variable. Also as growth was negative in some instances or estimates, this casts doubt on size class analyses which assume smooth monotonic growth.

Validity of the findings

The data are too weak and sparse to be statistically sound.
There is too much pointless repetition of previous studies on the same plant in the same basic area.

Additional comments

I found this article to consider too many topics but with insufficient detail.

·

Basic reporting

Is it correct to use llareta un-italizesed/common name for this species? Examples of this on page 4 line 96, line 113 and line 153 on page 6. Otherwise no comments.

Experimental design

No comments

Validity of the findings

No comments

Additional comments

I found the manuscript a pleasure to read and it related the character of this research project very well (it almost feels if I was on the field trip as well!). I think the piece is suited very well to Peer J.

·

Basic reporting

l.37-38: It is not clear if the size reported here (as the largest cushions) is for an unique individual, or for the merging of several individuals, if the last is the case, A. selago has similar "carpets" of cushions maybe larger than the reported here (10m) anyway a citation is needed to support this affirmation. About this subject, authors include in their discussion the potential of A. selago to make big multi-individual carpets, I don’t understand why it is in the discussion and not here. (Cerfonteyn et al 2011 could be useful)

l.95: There is a closing parenthesis, but not an opening one

l.111: R version needs to be reported and should also be cited, see citation() in your R session.

l.134. Authors said that small size classes are the most common along the three elevational ranges, but the mean area reported is 29m2, since from the histogram (Fig.3) is possible to deduce that no cushions larger than 15m2 were present at the two first ranges, is difficult to believe the reported mean area of 29m2, Should it be 2.9?

l.145. The same result should not be reported in two manners (fig. and table). Authors might choose one of them to report the growth rates. For me the table is more informative and easy to follow, moreover, a plot of points connected by lines seems to me not the most appropriate for this results; besides to share the year of measurement, individual plants are not part of a progressive measurement, so they should not need to be connected by lines. Instead, bars could be more adequate.

l.157-159 It is reasonable to propose that older individuals tend to allocate more biomass to vertical growth, also the water captation explanation is logic. But since this vertical growth was not measured in older individuals (in relation with young ones), for me it is a bit speculative to say that this water captation allow for increased growth rates.

l.163 Instead to report the "north" as a mean value, please include the numeric value of the mean.

l.238 Since (at least) the northern populations of A. madreporica and L. acaulis show also high volumes, authors can expand this suggestion to all cushion plants with high volumes and irregular shapes, which brings more generalization (and value) to the suggestion.

l.245-247 Despite interesting, Polylepis was not part of this work, nor a detailed methodology was proposed for its measuring, please delete this lines.

l.249 "...at the base a moderate..." should an "of" be needed before "a" in this sentence?

l.271 The authors' conclusions leave out the orientation results, which are the most interesting from my point of view. Are certain individuals growing slower for being in a "not so efficient" orientation? Is this orientation more variable on small (young) individuals than in older ones? Discussion about these simply questions could improve this section.

In the discussion, particularly in the ORIENTATION section, it will be prefered that the authors discuse first their present results (l261-269), prior to analyze previous works (l.249-260)

Experimental design

l.74-75: If would be useful to know the distance between transects along the altitudinal gradient.

It is not clear the number of cushions used in the orientation measurements, are the 406 individuals part of this analysis?

Why did the authors use three elevational ranges?

A more detailed description of bootstraping procedures should be included, it was made with replacement? without it? what was the n f the subsample used in each bootstrap iteration?

Despite the proportion of cushions with a north orientation seems to be important and suggest a trend, an statistical analysis would be necessary, the Rayleigh test for circular analysis can be appropriate here. Check the "circular" package in R.

Validity of the findings

No Comments

Additional comments

The present paper is an actualization of a long term survey about Azorella compacta, its growing patterns and population structure. This kind of plants are difficult to study because of their low growing rates and extreme high altitudinal environment, which confers particular value to any research related to this kind of species.

The article is well written, the authors have previous experience in this system and particularly, on this species, in this sense, their expertise and theoric background is reflected in the paper. However, some points need to be clarified and some others improved in order to produce an appropriate “unit of publication” as the journal standards suggest, and not just an actualization of some aspects of this species.

For achieve this, authors should include in the introduction a small paragraph of the previous findings that they have found, the authors do it in a way that a reader should search their previous papers (Kleier & Rundel 2004, 2009) between lines 39-44, in the same number of lines, authors could report the main parameters of the population structure (some of this is in the abstract), the energy requirements encountered and the importance of the orientation for the energy balance, which is for me the main result of this paper.

My other "conflict" is related to the "population dynamics" term, used by the authors even in the title of the paper. For me, a study about "population dynamics" should include analysis involving transition probabilities between ages, stages or classes, in this case the term "STRUCTURE of the population" seems more appropriate to the reported data. Actually the same authors use that term to discuss their findings (lines 183-209).

---

## Round 0.2 · Minor Revisions

Thank you for your revision. The paper has been substantially improved and tightened. I do, however, feel that the "why" of the paper is still not clearly articulated. For example, on line 217 of the results/discussion (of the tracked changes version), it is suggested that these plants are not good indicators of climate change (please check the sentence on 216-217, as it does not read well), however the idea of these plants as potential indicators of climate change is not clearly articulated in the introduction. On line 66 it is mentioned that sampling took place to "determine the presence of correlations of density or size with elevation"; again - the "why" is not articulated. If the authors can sell the value of reporting such data, it would strengthen the paper's message substantially.

Line 170. What does "these categories" refer to? No earlier mention of categories.
Line 243. what is meant by "radial growth from shrinkage"; growth and shrinkage are two different things? Please rephrase for clarity.
Line 264. replace "is not" with "does not occur at"
Line 306/307. The sentence starting "This orientation" is not well articulated.
Line 302. Don't start a paragraph with "However".

There are a few smaller editorial issues through the revision, so please give it a final thorough edit before re-submission.

---

## Author Rebuttal · Round 0.2

February 9, 2015

Dr. Karen Esler
Academic Editor for PeerJ
RE: Rebuttal Letter

Dear Dr. Esler,

Thank you for the opportunity to improve our manuscript. In this letter, we will address each point from the three reviewers.

**Reviewer 1**

Basic reporting

Reviewer 1: I feel the article is not coherent; it has bits on orientation, demography, elevational range and growth making it inappropriately subdivided.

> Our response: The reviewer comments that the article is not coherent. While we agree that we measured three different aspects of *Azorella compacta*, we feel that these aspects are related in determining the growth patterns of this unique species. Size class patterns are directly related to growth, which is directly related to orientation. We specifically changed the language in the introduction to reflect on previous work and tie these measurements together more coherently (lines: 46 – 51).

Experimental design

Reviewer 1: The core of the paper concerns growth estimates. Although 100 plants were marked only 9 were found. This is too few, certainly for 10 000 replications in a boot-strap analysis because growth was highly variable. Also as growth was negative in some instances or estimates, this casts doubt on size class analyses which assume smooth monotonic growth.

> Our response: The reviewer comments that it was inappropriate to use boot-strap analysis. We changed our statistical analysis from bootstrap to Wilcoxan Signed Rank test (lines: 141-142). We also included the Rayleigh test of uniformity (as suggested by reviewer 3).

Validity of findings

Reviewer 1: The data are too weak and sparse to be statistically sound. There is too much pointless repetition of previous studies on the same plant in the same basic area.

> Our response: We find that with the inclusion of the above language and the Rayleigh test on orientation that these findings are statistically sound. Additionally, the size class data are statistically significant, showing a clear trend toward many more smaller plants

and no clear trend with size and elevation.  This latter point is important as it shows *Azorella compacta* is not responding to climate change.   We don't find this to be "pointless repetition."

## Reviewer 2

Basic reporting

Reviewer 2:  Is it correct to use llareta un-italizesed/common name for this species? Examples of this on page 4 line 96, line 113 and line 153 on page 6. Otherwise no comments.

> Our response:  We removed the use of *llareta* throughout the paper, except for the one instance where we do report the common name, and we've italicized it.

## Reviewer 3

Basic Reporting

Reviewer 3:  It is not clear if the size reported here (as the largest cushions) is for an unique individual, or for the merging of several individuals, if the last is the case, *A. selago* has similar "carpets" of cushions maybe larger than the reported here (10m) anyway a citation is needed to support this affirmation. About this subject, authors include in their discussion the potential of *A. selago* to make big multi-individual carpets, I don't understand why it is in the discussion and not here. (Cerfonteyn et al 2011 could be useful).

> Our response: We removed the statement about *A. compacta* being the largest cushions and references to *A. selago* in the Introduction.  We also added the Cerfonteyn reference.

Reviewer 3: l.95: There is a closing parenthesis, but not an opening one

> Our response:  We corrected this error.

Reviewer 3 l.111: R version needs to be reported and should also be cited, see citation() in your R session.

> Our response:  We reported the version of R that we used in line144.

Reviewer 3:  l.134. Authors said that small size classes are the most common along the three elevational ranges, but the mean area reported is 29m2, since from the histogram (Fig.3) is possible to deduce that no cushions larger than 15m2 were present at the two first ranges, is difficult to believe the reported mean area of 29m2, Should it be 2.9?

Our response: The reviewer was correct that $29m^2$ should be 2.9 $m^2$, and this change is in line 153.

Reviewer 3: l.145. The same result should not be reported in two manners (fig. and table). Authors might choose one of them to report the growth rates. For me the table is more informative and easy to follow, moreover, a plot of points connected by lines seems to me not the most appropriate for this results; besides to share the year of measurement, individual plants are not part of a progressive measurement, so they should not need to be connected by lines. Instead, bars could be more adequate.

Our response: As per the reviewer's suggestion to not present data in both a table and a figure, we removed figure 4.

Reviewer 3: l.157-159 It is reasonable to propose that older individuals tend to allocate more biomass to vertical growth, also the water captation explanation is logic. But since this vertical growth was not measured in older individuals (in relation with young ones), for me it is a bit speculative to say that this water captation allow for increased growth rates.

Our response: We removed the text about water capitation that the reviewer found speculative. This was in line 157-159.

Reviewer 3: l.163 Instead to report the "north" as a mean value, please include the numeric value of the mean.

Our response: We rewrote the text regarding orientation and provided numeric values, lines 250-256.

Reviewer 3: l.238 Since (at least) the northern populations of *A. madreporica* and *L. acaulis* show also high volumes, authors can expand this suggestion to all cushion plants with high volumes and irregular shapes, which brings more generalization (and value) to the suggestion.

Our response: We rewrote this section to expand the suggestion.

Reviewer 3: l.245-247 Despite interesting, *Polylepis* was not part of this work, nor a detailed methodology was proposed for its measuring, please delete this lines.

Our response: We removed the text regarding *Polylepis*. These were lines 245-247.

Reviewer 3: l.249 "...at the base a moderate..." should an "of" be needed before "a" in this sentence?

Our response: We added the "of."

Reviewer 3: l.271 The authors' conclusions leave out the orientation results, which are the most interesting from my point of view. Are certain individuals growing slower for being in a "not so

efficient" orientation? Is this orientation more variable on small (young) individuals than in older ones? Discussion about these simply questions could improve this section.

Our response:  As per the reviewer's suggestion, we rewrote the conclusion.

Reviewer 3:  In the discussion, particularly in the ORIENTATION section, it will be preferred that the authors discuss first their present results (l261-269), prior to analyze previous works (l.249-260)

Our response:  We rewrote the orientation section to include the present study before citing other work.

Experimental Design

Reviewer 3:  It would be useful to know the distance between transects along the altitudinal gradient.

Our response:  We added that transects were at least 500 m apart, lines 88 and 89.

Reviewer 3:  It is not clear the number of cushions used in the orientation measurements, are the 406 individuals part of this analysis?

Our response:  We included the sample size for these data (n = 53), and included in line 140.

Reviewer 3:  Why did the authors use three elevational ranges?

Our response:  We used three elevational categories so the data could be better visualized, and we explained this on lines 170-171.

Reviewer 3:  A more detailed description of bootstraping procedures should be included, it was made with replacement? without it? what was the n f the subsample used in each bootstrap iteration?

Our response:  We changed the statistical analysis to a Wilcoxan signed rank test; therefore, we omitted the explanation of bootstrapping.

Reviewer 3:  Despite the proportion of cushions with a north orientation seems to be important and suggest a trend, an statistical analysis would be necessary, the Rayleigh test for circular analysis can be appropriate here. Check the "circular" package in R.

Our response:  We did use the Rayleigh test for uniformity in R, and this is stated in the text in lines 282-284.  We also included a new figure, Fig. 5 to illustrate this concept.

Validity of the Findings

Reviewer 3:  No comments

Comments for the author

Reviewer 3:  For achieve this, authors should include in the introduction a small paragraph of the previous findings that they have found, the authors do it in a way that a reader should search their previous papers (Kleier & Rundel 2004, 2009) between lines 39-44, in the same number of lines, authors could report the main parameters of the population structure (some of this is in the abstract), the energy requirements encountered and the importance of the orientation for the energy balance, which is for me the main result of this paper.

> Our response:  we rewrote the second paragraph of the introduction to reflect this comment, lines 42-58 (track changes) and lines 41-51 (new draft).

Reviewer 3:  My other "conflict" is related to the "population dynamics" term, used by the authors even in the title of the paper. For me, a study about "population dynamics" should include analysis involving transition probabilities between ages, stages or classes, in this case the term "STRUCTURE of the population" seems more appropriate to the reported data. Actually the same authors use that term to discuss their findings (lines 183-209).

> Our response:  we changed the term population dynamics to size class structure throughout the paper.

We sincerely hope these changes merit our manuscript worthy of publication, and again, we appreciate the reviewer comments.

All the best,

Cath Kleier (on behalf of all the authors)
Chair, Department of Biology, D-8
Regis University
Denver, CO  80212 USA

---

## Round 0.3 · accepted · Accept

Thank you for your corrections. Please note that on line 213, "travles" is spelt incorrectly, and will need updating in the proofs.